# Between Restrictive and Supportive Devices in the Context of Physical Restraints: Findings from a Large Mixed-Method Study Design

**DOI:** 10.3390/ijerph182312764

**Published:** 2021-12-03

**Authors:** Alvisa Palese, Jessica Longhini, Angela Businarolo, Tiziana Piccin, Giuliana Pitacco, Livia Bicego

**Affiliations:** 1Department of Medical Sciences, University of Udine, 33100 Udine, Italy; Jessica.longhini@uniud.it (J.L.); direzione.scientifica@burlo.trieste.it (A.B.); piccin.tiziana@spes.uniud.it (T.P.); 2Ethics Management for Clinical Practice Area, Azienda Sanitaria Universitaria Giuliano Isontina, 34149 Trieste, Italy; giuliana.pitacco@asugi.sanita.fvg.it; 3IRCCS Burlo Garofolo, 34137 Trieste, Italy; livia.bicego@burlo.trieste.it

**Keywords:** hospital, long-term care, physical restraint, physical device, nursing home, nursing

## Abstract

Physical restraints are still a common problem across healthcare settings: they are triggered by patient-related factors, nurses, and context-related factors. However, the role of some devices (e.g., bed rails), and those applied according to relatives’/patients’ requests have been little investigated to date. A mixed-method study in 2018, according to the Good Reporting of a Mixed Methods Study criteria was performed. In the quantitative phase, patients with one or more physical restraint(s) as detected through observation of a single index day in 37 Italian facilities (27 long-term, 10 hospital units, =4562 patients) were identified. Then, for each patient with one or more restraint(s), the nurse responsible was interviewed to gather purposes and reasons for physical restraints use. A thematic analysis of the narratives was conducted to (a) clarify the decision-making framework that had been used and (b) to assess the differences, if any, between hospital and long-term settings. The categories ‘Restrictive’ and ‘Supportive’ devices aimed at ‘Preventing risks’ and at ‘Promoting support’, respectively, have emerged. Reasons triggering ‘restrictive devices’ involved patients’ risks, the health professionals’ and/or the relatives’ concerns. In contrast, the ‘supportive’ ones were triggered by patients’ problems/needs. ‘Restrictive’ and ‘Supportive’ devices were applied based on the decision of the team or through a process of shared decision-making involving relatives and patients. According to the framework that emerged, long-term care patients are at increased risk of being treated with ‘restrictive devices’ (Odds Ratio 1.87, Confidence Interval 95% 1.44; 2.43; *p* < 0.001) as compared to those hospitalized. This study contributes to the improvement in knowledge of the definition, classification and measurement of physical devices across settings.

## 1. Introduction

‘Physical restraint’ has been defined as any measure aimed at controlling a patient’s physical movement that cannot be easily removed by the patient him/herself, including bed rails, limb or abdominal belts, fixed tables and straitjackets [1]. Over the past 20 years, alongside the ongoing ethical debate, several studies [2,3] have demonstrated both the direct and indirect negative effects of physical restraints. Pressure ulcers, strangulation, muscle, nerve, or vascular injuries, decreased mobility, falls, physiological illness, and prolonged length of stay have all been reported as consequences, with some studies also documenting a direct association with death [3,4]. Although available evidence has emphasised the lack of effectiveness in promoting safety, such as in avoiding falls [5], the physical restraint is still widely used mainly for safety purposes [6]. Prevalence data in non-psychiatric hospital settings have been reported to range from 1.6% in medical-surgical [7] to 8.3% in medical units [4], and up to 59% in intensive care units [8]. High occurrence has also been reported in nursing homes and in intermediate and long-term settings, where physical restraint prevalence has been documented to range between 17.4% [9] and 96.6% [10]. The wide variability across both different and similar units has been mainly attributed to the lack of an operational definition on how to measure the occurrence of physical restraint use [11].

In order to identify triggers and alternative interventions, a growing body of research has investigated the underlying nurses’ decision-making process leading to the use of a physical restraint [12,13]. Different tools have also been validated, such as the Staff Knowledge, Attitudes, and Predictors Questionnaire [14] and the Physical Restraint Use Questionnaire [15]. In these tools, there are assessed a list of predefined reasons of restraints use, the possessed knowledge, and the attitudes of nurses. According to the evidence available, decisional triggers of physical restraint use have been categorised in: (a) patient-related factors, such as risks of falls or device removal, and aggressive behaviours; (b) context-related factors, such as the lack of human resources and the work environment quality [16]; and (c) nurse-related factors, regarding the education, expertise and attitudes. The inadequate education and the negative attitudes have been reported as increasing the use of physical restraints [17]. In line with this, targeted educational interventions have been documented to be effective in preventing physical restraint use [18]. 

To date, the majority of studies have investigated physical restraints, nurses’ knowledge, attitudes, beliefs and underlying reasons as prevalence studies [19,20,21]. In contrast, only a few studies have attempted to establish the decision-making process frameworks leading to the physical restraints use, and those available have considered nurses as the main decision-makers [22]. Additionally, relatives’ point of view has been mainly investigated as in agreement or disagreement with nurses’ decisions, and patients’ experiences have been little explored, predominantly in psychiatric care [23]. However, the sparse available data on patients/relatives’ requests to apply some devices (e.g., bed rails) [24] requires reflection on what physical restraint is [25], especially among older and frail patients who are at high risk of being restrained [4]. The same physical device, such as a bedrail, might function as a restraint (e.g., among patients with delirium) and also as a therapeutic intervention when requested by a patient in need to move his/herself independently in the bed [26]. As a consequence, despite a considerable amount of literature [3], controversial considerations are still present regarding some physical restraints. For example, bed rails [27] have been defined as ‘moderate restrictive measures’ in some cases [28], thus underlining that restraints are not homogeneous, even when involving the same ethical principles.

Therefore, to expand the available conceptual definitions established through consensus [1], there is the need of an explicit operational definition and more research on the underlying decision-making process leading to the use of physical restraints [25,29]. Improving the available knowledge in this field might have multiple benefits. For example, it might support healthcare professionals, especially nurses, in their decision-making process and it might also support researchers in collecting accurate data preventing the under/overestimation of the phenomenon [21]. Moreover, it might help in investigating variations across settings according to the underlying reasons for physical restraint use. As last, increased knowledge on this topic might open new lines of ethical, professional, and methodological investigation. Thus, this study is aimed at contributing to the advancement of knowledge on the operational definition of physical restraints based on the nature of the reasons and purposes that lead to the decision-making process in daily practice.

## 2. Materials and Methods

### 2.1. Study Aims and Design

The following research questions were addressed: (a) what are the underlying reasons and purposes triggering the nurses’ decision to apply one or more physical restraint(s) in daily care practice? and (b) according to the findings that emerge, are there any differences across settings concerning the reasons underlying this decision?

An explanatory sequential mixed-method study design [30] was performed in 2018 with the purpose of deeply explaining the occurrence of physical restraints by investigating the complexity of the phenomenon in the light of the nurses’ voice, process of thinking and daily experience. Considering the three elements and methods proposed by Creswell and Clark [30]—sequence, priority, and integration—this mixed-method study was based on the following characteristics: (a)Sequence: (1) the quantitative method (=cross sectional study) was used as the first phase to collect the occurrence of physical restraints; (2) a descriptive qualitative method was used as the second phase to collect narratives on the underlying reasons for each device used to develop a decision-making framework; then, (3) a quantitative method was used to analyse data emerging from the both previous quantitative and qualitative phases;(b)Priority: the qualitative method prevailed in terms of weight throughout the research project to understand the nurses decision-making process to a larger extent (‘QUAL→quan’, [31]).

Integration: data integration was performed with multiple purposes [32,33]. Firstly, data collected from the first phase were used to identify the sample [34,35] of the second phase (qualitative descriptive study). Then, findings collected in the first phase and in the second phases were combined with the purpose of expansion and enhancement of the findings [32,33,34,35]. In order to develop and report the study here, the Good Reporting of a Mixed Methods Study (GRAMMS [36]) was followed (Appendix A).

### 2.2. Setting and Sample

The study was conducted in 37 facilities in Friuli Venezia Giulia (Italy), of which 27 were long-term care settings and 10 were hospital units. No specific guidelines or protocols for physical restraint use in these settings were available at the time of the study. At the time of the study, all settings involved the conventional bedrails positioned along the entire edge of the bed were used and applied in a fixed mode. Moreover, nurses were allowed alone to apply physical restraints as an extraordinary intervention under the medical prescription or after an in-depth assessment of the patient needs (Italian Nurses’ Deontological Code [37]). As defined in the Italian Penal Code, the use of restraints can be legitimated by the necessity situation, defined as an actual risk of danger for the patients or others. However, the adoption of restraints, considering the limitation of the freedom right, can turn in misdemeanour for violence, kidnapping, and abuse of correction measures.

In the first quantitative phase, the target population was patients and residents: all patients who were present in the hospitals or long-term units on a single day were included. Patients were excluded if they were not present on the units (e.g., absent for medical visits). Therefore, a total of 4562 residents and patients (hereinafter, patients), 3933 of them living in long-term facilities and 629 hospitalised, were included.

In the second qualitative phase, the target population was all nurses in charge of one or more patient(s) physically restrained and willing to participate. Nurses were excluded if they were not involved in the care of the restrained patients and if they refused to participate in the study. A total of 90 nurses responsible for the care of restrained patients (66 nurses in the long-term facilities and 24 nurses in the hospital units) participated.

### 2.3. Data Collection

In the first quantitative phase of the study, data regarding the total number of patients cared for on a single day and, among them, who reported one or more physical restraint(s), were collected. To avoid under- or over- reporting of the physical restraint use, several strategies have been adopted. A total of 36 involved researchers were trained in a one-day course aimed at standardising the data collection process. Restraint devices were collected by using the list of devices included in the Maastricht Attitude Questionnaire data collection tool [28], after having obtained formal authorisation (Prof. Mayer/Hamers, correspondence available from authors). The list of devices was translated (backward–forward procedure) (Table 1), and piloted both in long-term facilities and in hospitals. According to its checklist nature, no further validation measures were assessed. Researchers accessed the randomly assigned units on one selected morning, which was also randomized. For each unit, there were identified two researchers that were not involved in the care of patients. Their approach was not intrusive; one of them approached patients while wearing a nurse uniform and presenting her/himself. Then, after having obtained the consent from the patient, his/her body was inspected, and all physical restraints were recorded on the form (Table 1). Moreover, in the case of patients not able to give their consent (e.g., with dementia), researchers collected data through direct observation during daily activities. The data collection at the bedside ranged from one hour to 4 h, according to the size of the long-term facility/unit.

In the qualitative phase, nurses in charge of patients with physical restraints were interviewed by the second trained nurse researcher. The eligible nurses were approached in each facility, just one hour after the quantitative phase; each of them was asked to participate and no nurse refused the interview. They were reminded of the study aims and then, in a private room of the facility—when the eligible nurses were available—an individual semi-structured interview was performed focused on the following questions: ‘This morning, Mr/Mrs … was physically restrained: what were the main purpose and reason(s) leading to the decision-making process in applying physical restraints in this patient/resident?’. According to the nurse-to-patient ratio, each nurse was interviewed on 10 (hospital units) to 27 (long-term facilities) patients with physical restraints, and thus the duration of the interview varied from 10/15 min to around 40 or more minutes. Nurses’ answers were recorded and transcribed verbatim. No data on demographic characteristics for nurses have been collected in order to ensure narration freedom, given the peculiar nature of the topic implying emotional, legal and ethical issues and dilemmas [37,38].

### 2.4. Data Analysis

Firstly, a thematic analysis [39] was performed using an inductive approach [40]: each reason mentioned by nurses was analysed and compared with by three researchers (see authors) independently; these reasons were then classified. When a request by a patient or relative emerged in the nurse’s narration, the misinterpretation was avoided by documenting the exact words reported by the patient or relative to the nurse as recalled by him/her. Furthermore, to ensure trustworthiness, the emerging decision-making framework was validated with two different strategies: (a) first, two researchers not involved in the thematic analysis (see authors), classified again the emerged reasons by re-performing the entire process independently; (b) second, the emerged framework was discussed in two meetings involving all participant nurses, their chief nurses and other representatives of the facilities/hospitals (namely, 140 in the first and 30 in the second meeting) thus performing a member-checking [41].

After having developed the decision-making framework, data were transformed in numeric values by counting frequencies and percentages; then descriptive and inferential analysis was performed in order to identify the differences, if any, in the occurrence of the reasons between long-term and hospital settings. The same researchers involved in the first phase conducted the integration process of the data. Specifically, the number of patients cared for, the number of them with physical restraint(s) and the reasons were reported as absolute and percentage frequencies. To address the second research question, inferential analysis was used to test the hypothesis if the setting, as the exposure independent variable, was associated with the status of being restrained and to be restrained with for a specific reason, as the dependent variable. Differences in the dependent variables between long-term and hospital facilities have been assessed using the chi-squared test (χ^2^) and by calculating the Odds Ratio (OR; confidence of interval, CI 95%). The statistical difference was set at *p* < 0.05 and the analysis was performed using the SPSS Statistical Package Version 25.

### 2.5. Ethical Issues

The research project named CONT_EXIT (‘A strategy to exit from a routine use of physical restraints’) was approved by the regional Ethical Committee (24653/CER, 2017). The project was then presented in all facilities and hospitals through meetings and also through the media in order to inform citizens. Several strategies were adopted in order to address the ethical issues during data collection:(a)patients’ dignity was ensured by requiring a verbal permission to assess the presence of physical restraint measures; in those not capable of expressing their consent, researchers observed the residents during their care activities to discover the presence/absence of the restraint(s);(b)the nurses involved participated on a voluntary basis and consent was collected; moreover, individual interviews were conducted in a dedicated space with no other people present or in transition and without installed video camera. Additionally, demographic and sensitive data of nurses was not collected to ensure anonymity, privacy, confidentiality, and freedom in narrations; an identification code, not linked to sensitive information, was assigned to each nurse;(c)the settings were also anonymised, and data analyses were performed including all hospital and all long-term facility units, in order to ensure confidentiality.(d)However, a detailed description of the findings that emerged in each unit was returned to the chief nurse in order to promote an internal critical reflection and the identification of strategies for improvements

## 3. Results

### 3.1. Reasons and Purposes Triggering the Decision to Apply Physical Restraint(s)

Of the 4562 patients, 2233 (48.9%) were physically restrained due to 14 main reasons that have been categorised into two main purposes, ‘Preventing risks’ and ‘Promoting support’.

Devices ‘Preventing risks’ were aimed at preventing negative and/or adverse events (e.g., falls), and were triggered by the following reasons:patients’ actual clinical problems, such as confusion/disorientation (=bewilderment, emotional disturbance, lack of clear thinking, and perceptual disorientation), agitation (=such as the patient reporting restlessness associated with increased motor activity), or wandering (=moving through space while confused or otherwise cognitively impaired);patient uncertainty (e.g., fluctuant, not measurable), decreased or absent risk awareness, resulting in being incapable of protecting him/herself;patients’ assessed risks leading to fall(s), device self-removal or self-harm;patients’ safety needs to be protected from complications (e.g., accidental device removal, such as a urinary catheter) or from additional issues/diseases accidentally acquired (e.g., preventing infections due to the accidental touch of a surgical site);healthcare professionals’ safety needs, such as protection from patients’ aggressive behaviours; andrelatives’ concerns/needs, triggering a formal request to nurses to apply a physical restraint, mainly bed rails, during their absence.

On the other hand, devices triggered by the intent of ‘Promoting support’, were grounded on reasons aimed at promoting positive outcomes such as: (a) improving comfort (e.g., positioning bed rails); (b) sustaining/promoting posture (e.g., fixed-tables in chair sitting) in patients not capable of maintaining a healthy posture due to their health conditions, or (c) enabling their mobilization (e.g., positioning bed rails to ensure patients’ ability to turn in bed autonomously); and (d) meeting the patients’ needs, for example, to feel protected. 

Therefore, those devices intended for ‘Preventing risks’ have been defined as ‘Restrictive device’ (in other words, ‘true restraints’), while those beneficial devices intended for ‘Promoting support’ have been defined as ‘Supportive devices’, as summarised in Figure 1. 

The decision-making process to apply a physical device has emerged as being led by the healthcare professional team according to patients’ problems, risks or needs and the safety needs of the healthcare professionals, whereas shared decisions included those in which the team undertake the decision according to the needs expressed by relatives or by patients. 

### 3.2. Differences across Settings

A total of 1965 patients out of 3933 (49.9%) and 268 out of 629 (42.6%) reported one or more physical restraint, significantly more in long-term facilities as compared to the hospital units (*p* < 0.001). By taking into consideration the framework emerged (Figure 1), in long-term facilities, 27.1% of patients were restrained with ‘Restrictive devices’ and 22.8% with ‘Supportive devices’. Differently, an opposite trend emerged in hospital units where 16.5% of patients were restrained with ‘Restrictive devices’ and 26.1% with ‘Supportive devices’ (*p* < 0.001) (Table 2). Therefore, long-term care patients were more likely to have a device (OR 1.34, CI 95% 1.13; 1.59, *p* < 0.001) in general, and a ‘Restrictive device’ (OR 1.87, CI 95% 1.44; 2.43; *p* < 0.001) as compared to hospitalised patients.

As reported in Table 2, no statistical differences in the decision-makers between settings emerged regarding the use of ‘Restrictive devices’ (*p* = 0.07) and ‘Supportive devices’ (*p* = 0.62).

Confusion and disorientation triggered significantly more often the ‘Restrictive devices’ among hospitalised patients compared to those living in long-term facilities (27.9% vs. 6.8%, *p* < 0.001). Similarly, healthcare professional safety needs triggered ‘Restrictive devices’ more often among hospital patients compared to those living in long-term care settings (16.3% vs. 3.0%, *p* < 0.001). 

In the context of ‘Supportive devices’, sustaining posture was the reason more significantly reported for patients in long-term care settings as compared to those hospitalised (72.2% vs. 48.8%, *p* < 0.001). Furthermore, enabling mobilisation was more often reported for hospital patients as compared to those living in long-term facilities (34.8% vs. 13.1%, *p* < 0.001). No differences have emerged for the remaining reasons (Table 2).

## 4. Discussion

We performed a mixed-method study in attempt to clarify the decision-making process underlying each physical restraint seen in the real word of practice. By considering the narrations of nurses, we developed a practice-grounded decision-making process framework, to advance the operational definitions available, and to discover differences across care settings. No previous studies in this field of research have been performed by involving a mixed-method design. Moreover, we accessed several settings (*n* = 37), more than those on average included in available studies, e.g., [21], mixed in their mission, such as long-term facilities and hospitals, while current literature in the field has differentiated these settings to produce different research lines, e.g., [8,38]. Moreover, we involved 4562 patients, which is also higher than the average number documented in available studies, e.g., [39]. This suggests that our findings might provide suggestions on how the decision-making process regarding the physical restraint(s) use is undertaken on a daily basis in different settings for patients with different clinical conditions.

### 4.1. Restrictive and Supportive Devices

Not all physical restraints we have seen applied to patients were ‘restrictive’ restraints: the same device might function as a ‘restrictive’ or a ‘supportive’ device. According to the findings emerged, a ‘Restrictive device’ is defined as those devices limiting movement in order to prevent risks, in line with available definitions [1]. A ‘Supportive device’, on the other hand, can be defined as beneficial ‘devices’, aimed at supporting posture, or at enabling movements and at increasing comfort, thus promoting the patient’s independence and comfort. While preventing risks has already been reported as the main intent of physical restraints, e.g., [20], ‘promoting support’ seems to be a new concept in this research field given also the limited investigation of the reasons underlying the so-called ‘physical restraint devices’ use [42]. This new finding has emerged from our study given the combination of quantitative and qualitative approaches. Moreover, participant nurses have been stimulated to reflect on a given device including relatives or patient requests, rather than to explain or justify their behaviour/decision by contextual factors and work environment features (e.g., poor nurse-to-patient ratio, absence of family, lack of knowledge) [20,21]. Additionally, in order to achieve the aims of the study, observations, interviews and data analyses were guided by researchers trained on physical restraint issues and with clinical experience in different settings, thus ensuring the expertise required in evaluating the phenomenon and in addressing all ethical issues during the study phases. However, collecting more data on clinical experience and knowledge of both participant nurses and researchers would be helpful to further explain the findings given. Nurse expertise and education have been demonstrated as affecting physical restraint occurrence [17,43], as well as the validity of the qualitative data codification performed.

Differentiating physical restraints according to their intent into restrictive and supportive device categories might have several potential implications:first, this classification differentiating between positive beneficial and negative devices suggests that the accurate measurement of the physical restraints use should not be reduced to the counting of the presence of a device or to the evaluation of movement limitation; therefore, alongside direct observations, the underlying intents and reasons should also be assessed in future studies. Moreover, assessing the intent might increase the accuracy in reporting some devices that apparently might function both as a restraint and as a support (e.g., bed rails, [44]);second, this classification might overcome available definitions reported in some studies, as in the case of bed rails that have been considered as ‘light restraints’ [13], as it would be possible to differentiate the degree of restraints as ‘light’ or ‘heavy’ according to the degree of freedom available with the device;third, classifying restraints as restrictive and supportive devices might help in appropriately considering those applied according to the patients’ request that have already been discussed based on their meaning (e.g., bed rails, [24]) but not truly legitimated in the debate as appropriate restraints or not;fourth, better differentiating the concepts might help inform educators and researchers in an understanding of how to further limit the physical restraint use, targeting the training on those avoidable and illegal restraints such as those used with the purpose of preventing risks.

Thus, the emerged findings suggest that ‘restrictive device’ might be referred to and reported in patient records as ‘physical restraints’: strategies aimed at limiting or preventing their use are strongly recommended. On the other hand, those devices enabling patients according to their needs might be considered ‘supportive devices’ or ‘physical enablers’. To date, only Mullette and Zulkowski have recognised the role of bed rails as enablers [45].

The second main finding of this study concerns the decision-makers. To date, the decision-making process has been explored mainly according to the nurses’ and health professionals’ perspectives. Consequently, tools have been developed with the aim of investigating health professionals’ beliefs and subjective norms with regard to the intention to use physical restraints [20]. According to our findings, three players are involved in the decision process: (a) healthcare professionals, who decide to apply restraints according to a multidisciplinary decision, suggesting that no one profession seems to undertake this decision alone; (b) relatives; and (c) patients who are involved in a shared decision-making process. Alongside the professionals’ responsibility to not harm patients and to ensure advocacy, more consideration should be devoted to the patients’ and relatives’ perspectives and experiences. In our study, relatives and patients requested the application of physical devices as expressed by nurses. This suggests that the meaning of their needs and concerns should be investigated carefully and not merely as an agreement or disagreement with the team’s decision. In the few available qualitative studies, patients overall felt they were being helped and cared for when a physical restraint was applied; in this context, patients have been reported to have understood the rationale and the importance of preventing dangerous behaviours [46,47]. However, greater efforts in research are needed to explore in depth patients’ and relatives’ perspectives in this field.

The third main finding is that the major reason for using a physical device has also been reported in previous research [20] as patients’ problems and risks, including agitation, disorientation, fall risk, device-removal risk and self-harm. With regard to the patients’ safety issues, the nurses involved seem to have considered the Fundamentals of Care framework [48] where, among the discrete physical elements, patient safety is mentioned (e.g., risk assessment and management, infection prevention, minimising complications). However, other reasons have emerged, such as the need for healthcare professionals to perceive the situation as under control and safe—previously documented in available tools [15].

With regard to the relatives, while their role has been largely documented in home care [49], no trace of a similar role has been reported for long-term settings and only a few examples are available in hospital units [26]. In our study, ‘restrictive device’ decisions triggered by relatives might be interpreted in different ways. They might have perceived the high workloads of healthcare professionals, and with the aim of preventing risks at need of close surveillance, they might have requested the restraints. They might also have learnt about the intervention implemented by healthcare professionals by witnessing it and then simply asking to provide the same when they leave the unit. This latter also calls for the need to investigate in depth the role of relatives to better understand their real contribution in the occurrence of a ‘restrictive device’. For example, whether on the one hand they might be considered as being among those who increase their use by asking nurses to apply a restraint method, on the other hand, they might contribute to prevent the use of a ‘restrictive device’ with their presence. In fact, the engagement of relatives and more broadly ‘health carers’, including volunteers, has been reported as a strategy preventing the physical restraint occurrence [29,38].

Reasons that have triggered the use of ‘supportive devices’ were sustaining posture, promoting comfort, and enabling mobilisation, as well as answering the need of patients who feel themselves more protected. These reasons have never been documented before, with the exception of bed rails [27,42]. Thus, the findings suggest that some physical devices can have an extended role, not only to limit freedom, but also to potentiate the residual abilities and to promote independence. Reporting appropriately these devices in the clinical records with their underlying intent(s) might contribute to a more accurate measure of physical restraint occurrence in practice, but might also enrich the debate on the ethical implications.

### 4.2. Differences across Settings

At first glance, half of patients in long-term care settings were restrained, and 42.6% were restrained in hospitals, suggesting a higher prevalence of physical restraints compared to the available data [41]. However, other researchers [8,10] reported a higher occurrence in both settings as compared to our data. These variations across studies might be explained under different lines of reasoning. Firstly, at the time of our study, physical restraint policies were not implemented, and no projects preventing their use were established in the settings. Moreover, nurses were guided by the Italian Nurses’ Deontological Code [33]. Since 2019, norms and guidelines at the national, regional, and local levels have been implemented and also a revision of Italian Nurses’ Deontological Code [33] was established. Secondly, according to the findings, the restrictive devices amount to about half of the cases (27.1% in long-term facilities, 16.5% in hospitals), confirming that not all restraints applied are intended to limit the freedom of patients. ‘Supportive devices’ emerged as having a higher occurrence in hospital settings, mainly due to the intent to enable mobilisation as compared to long-term settings, although sustaining the posture was a reason significantly higher in long-term care. In this context, bed rails might help in sustaining, for example, a pillow placed to ensure the side position and to prevent pressure ulcers. The use of physical devices as enablers has also been introduced recently in a prevalence study reporting that functional dependency is a predictor of bed rails use in rehabilitation facilities [50].

As a consequence, our findings suggest that bed rails, as the device mostly used to help patients, might play a crucial role in the appropriate measurement of physical restraints use, influencing the data collected. In our study the prevalence (42.6%) emerged across hospitals is higher as compared to that documented previously from 8% [26] (8%) to 5% [51]. However, these two studies were conducted in a large sample of hospitals without considering bed rails as a physical restraint device. In particular, Minnick and colleagues [26] included data only when bed rails were used with another physical restraint method, explaining that bed rails can be used also for other purposes than restraint (e.g., helping movement). In their next study on full bed rails use, different to our study in which we have also included one side bed rail, the overall rate was 11 for 100 patients/day, with the highest rate among intensive care units (55 per 100 patients’ days, [42]). This suggests the main role of bed rails in affecting physical restraint measurements.

All these examples suggest that it is not possible to attribute an absolute negative or positive interpretation to each device and to the prevalence data according to the different operational definitions of physical device [25]. Differently, recording the device used, in addition to the underlying intent and reasons, might help to measure the phenomenon accurately, to compare findings documented in the literature, and to design appropriate interventions. For example, when the underlying reason is to promote the posture, other new designed non-restrictive tools such as chairs or beds should be used rather than bed rails. Furthermore, although we are trying to understand the phenomenon from a new point of view, risks should also be carefully considered when the purpose is about enabling patients. Bed rails might be harmful considering the risk for the patient to be injured during the movement or delirium episodes.

Patients in long-term care settings had a 34% and 87% higher risk of being restrained and restrained with a ‘restrictive devices’, respectively, as compared to patients in hospitals and this might be linked with the higher risk of falls as reported by nurses in long-term care settings. Moreover, in these settings, the low nurse–patient ratio might decrease the surveillance capacity of staff, thus triggering the use of physical restraint. However, preventing the risk of falls is also the most used reason reported by nurses working in hospitals, which is similar to previous research [52]. This finding revealed that, despite the large amount of literature demonstrating that physical restraints do not prevent falls [5], this evidence is still not translated into practice.

The decision of the team prevailed in both settings—however, decisions have also been reached by involving relatives (from 2% to 4.8% in long-term and hospital settings, respectively) and patients (13.2% and 14.6%), without any statistical differences between settings. This confirms that physical devices should also be considered in the wider context of shared decision-making and not only as the decision of nurses.

### 4.3. Study Limitations

This study is affected by several limitations: first, nurses have been interviewed for several patients, according to the nurse-to-patient ratio documented in Italy [53]. Thus, the accuracy of data reported regarding the reasons for restraint use in each patient, where the presence of one or more restraint(s) have been observed, might have been threatened by the number and the complexity of the patients cared for. Secondly, only the main reason and intent were asked, thus taking into consideration the main cue triggering the decision [54]; therefore, future studies might also consider the reasons according to the types and the number of devices applied. Third, no data have been collected regarding the features of the facility (e.g., size, mission, unit type), the profile of nurses, and that of restrained patients (e.g., clinical issues) given that the aim of the study was to develop a decision-making framework regardless of these variables. Fourth, methods were integrated by collecting quantitative and qualitative data in the same broad context, which might be influenced by the same culture regarding the physical restraints; moreover, we have involved only nurses. Therefore, future studies are encouraged to collect more data regarding the setting, the healthcare professionals, and the patients to further develop the decision-making framework according also to other complex variables that might affect the decision. Moreover, in order to detect the intent of the physical restraint as experienced by patients and/or relatives, data regarding the benefits and/or damages perceived by them is encouraged to be collected. Furthermore, given the recent changes in the Italian Deontological Code of Nurses [51], a continuing investigation of factors affecting the decision-making process is encouraged.

## 5. Conclusions

This study contributes to the improvement of knowledge on the definition, classification, and measurement of physical restraints across settings. Firstly, in daily practice, according to their intent, physical restraints as ‘restrictive devices’ aimed at limiting freedom should be differentiated from the ‘supportive devices’, or ‘physical enablers’ which are aimed at promoting the residual potentialities, the comfort and the freedom of patients. However, each type of device should be applied with a careful consideration of risks, balancing the benefits, regardless of the good intentions or patient and relative requests. Secondly, in future studies, physical device use should be classified as ‘team-decision’ and ‘shared-decision’, thus also valuing the perspectives and the needs of both relatives and patients and promoting a vision in which nurses are not alone in decision-making but are rather a component of a team. Furthermore, data regarding the intents of the devices applied should be collected in addition to the direct observation at the bedside in order to understand the real significance of the intervention applied to the patient. Finally, both in research projects and policy document development, the differences and features of settings should be considered to improve the effectiveness of specific interventions aimed at preventing the occurrence of ‘restrictive devices’.

## Figures and Tables

**Figure 1 ijerph-18-12764-f001:**
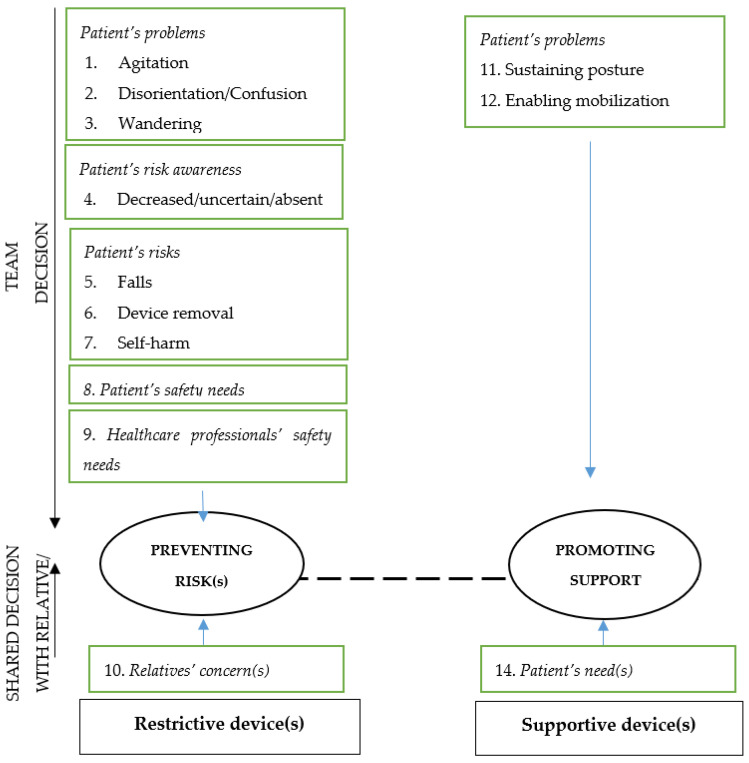
Decision-making framework.

**Table 1 ijerph-18-12764-t001:** Data collection grid: brief presentation [28].

Setting: Long-term care; Hospital (unit) Patient/Resident_____________(number) During direct observation, was this patient/resident restrained with a physical measure? Yes/No If yes, indicate one or more option in the list below. ▪ Both sides bed rails ▪ One side bed rail ▪ One side bed rail + the other side near the wall ▪ Fixed table on chair ▪ Containment pyjamas ▪ Unilateral upper-limb wrist belt ▪ Bilateral upper-limb wrist belt ▪ Unilateral lower-limb ankle belt ▪ Bilateral lower-limb ankle belt ▪ Abdominal belt in chair sitting ▪ Abdominal belt in bed lying ▪ Straitjacket ▪ Pelvic belt ▪ Tight sheet to limit rising ▪ Locked the door ▪ Other (please indicate)

**Table 2 ijerph-18-12764-t002:** Differences between settings regarding the nature, the decision-makers, and the reasons of physical devices.

	Long-Term Care *n* (%)	Hospital *n* (%)	*p*-Value
Total number of patients	3933	629	--
Patients restrained with restrictive or supportive device(s)	1965 (49.9)	268 (42.6)	<0.001 ***
**Patient with restrictive device(s), *n***	1066	104	
% Of total patients	(27.1)	(16.5)	<0.001 ***
% Of patients restrained	(54.4)	(38.8)	
Decision makers: according to			
Team decision	1045 (98)	99 (95.2)	0.07 ^†^
Shared decision with families and patients	21 (2)	5 (4.8)	0.07 ^†^
Reasons: Preventing risks			--
Agitation	127 (11.9)	10 (9.6)	0.48
Disorientation/confusion	72 (6.8)	29 (27.9)	<0.001 ***
Wandering	12 (1.1)	0 (0)	0.61 ^†^
Decreased/uncertain/absent risk awareness	26 (2.4)	2 (1.9)	0.74
Fall risk	704 (66)	32 (30.8)	<0.001 ***
Device-removal risk	6 (0.6)	3 (2.9)	<0.01 **
Self-harm risk	2 (0.2)	0 (0)	1 ^†^
Patient’s safety need	64 (6)	6 (5.8)	0.92
Healthcare professionals’ safety needs	32 (3)	17 (16.3)	<0.001 ***
Relatives’ concern(s)	21 (2)	5 (4.8)	0.07 ^†^
**Patient with supportive device(s), *n***	899	164	<0.001 ***
% Of total patients	(22.8)	(26.1)	
% Of patients restrained	(45.7)	(64.2)	
Decision makers: according to			
Team decision	780 (86.8)	140 (85.4)	0.62
Shared decision with families and patients	119 (13.2)	24 (14.6)	0.62
Reasons: Promoting support			--
Sustaining posture	649 (72.2)	80 (48.8)	<0.001 ***
Enabling mobilisation	118 (13.1)	57 (34.8)	<0.001 ***
Improving comfort	13 (1.5)	3 (1.8)	0.72 ^†^
Patient’s need(s)	119 (13.2)	24 (14.6)	0.63

Legend. N, Number; ** *p* < 0.01, *** *p* < 0.001; ^†^ Fisher’s Exact Test.

## Data Availability

Data available on request due to restrictions, e.g., privacy or ethical.

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
