# Peer review of "Between Restrictive and Supportive Devices in the Context of Physical Restraints: Findings from a Large Mixed-Method Study Design"

_ijerph, 2021, doi:10.3390/ijerph182312764_

Round 1

Reviewer 1 Report

Thank you for asking me to review this manuscript that reports on a very important and contemporary issue – physical restraint. While the topic and findings are of clinical significance, some of the writing needs to be reviewed as the clarity of the work is lost due to grammatical anomalies. There are also other aspects that could be considered in relation to the conclusions drawn.

Abstract: in the abstract and throughout the paper reference is made to the index day – this needs to be explained for the international audience. The sentence starting with ‘Restrictive’ and ‘Supportive’ devices…….. seems incomplete.

General writing – in sections, sentences are very long which makes it difficult for the reader to follow. An example is page 2 line 55-62. Suggest a full stop after ‘… expertise and attitudes. Specifically ……’. Also page 2 line 84-90. Suggest rewriting to remove the a) b)…. And write as shorter sentences.

Data collection: please explain what is meant by after having obtained formal authorisation and also Their approach was gentle

Please explain how consent was gathered for people with dementia.

Differences across care settings – this section is difficult to follow. Suggest rounding % up. Also the reporting of the % of restrictive devices and supportive devices is not clear. Is it possible to express this as a %total/care setting (out of 100%) – it took some time to work out what had been reported.

Page 7 line 302 – this is not a complete paragraph

As acknowledged in the limitations, not having the demographics reported of the patients is a significant issue with this manuscript and detracts from the generalisability of findings. If these are available it would be helpful to include.

The discussion is interesting. However, the use of very long sentences and tendency to make lists of points makes reading this tedious. Suggest reviewing this section with the aim of shortening would help

While I can see there is a need to define why a restraint is being applied, and this paper makes an important contribution, I have concerns related to the idea that there is a difference between the two types of restraint. Even though a bed rail is left up to help the person roll, there is still a risk that they could become trapped in the bedrail or become injured because the bedrail is up. I believe it is important for the authors to acknowledge the risk even to these people and to consider how this new evidence could inform guidelines for safe use of supportive restraint. Also are their new designs of chairs and beds that promote autonomy while providing support but are not restrictive. I would be concerned that the focus is that there are two types of restraint as this suggests the supportive type should be allowed without consideration of risk. Please consider these points in your discussion

Author Response

Thank you for the appreciation and valuable suggestions to improve the manuscript. Please see the attachment.

Reviewer 2 Report

Thank you for giving me the opportunity to review the manuscript, entitled “Between restrictive and supportive devices in the context of physical restrains: findings from a large mixed-method study design”. From the nursing perspective, the use of restraint should be considered as an intervention of last resort.  Overall, suggest to discuss the legal aspects of the use of physical restraint in Italy. Please see below some suggestions and comments regarding details of clarifications and ethical considerations.

  • On p.1 line 17, “….= 4562 patients) were identified; then, for each patient, the nurse responsible was interviewed…”. Clarification is needed since p3 line 133 mentioned that only 90 nurses has participated in the study. Suggest to review the numbers.
  • On p.1 line 21 “Reasons triggering ‘restrictive’ devices involved patients, healthcare professionals, and relative’s risk/needs/ concerns; in contrast….” Suggest to rewrite this part as “ Reasons triggering ‘restrictive devices’ involved patients’ risks, and the health professionals’ and/or the relatives’ concerns. In contrast….”
  • On p.1 line 39, suggest to change “…longer in hospital stay” to “prolonged hospital length-of-stay”.
  • On p.1 line 41 mentioned “… missed effectiveness of promoting safety”, suggest to discuss what does it mean by “missed effectiveness”. Consider when the restraint is used for the purpose of prevent fall risk, will this case be considered as “missed effectiveness”. Suggest to elaborate further.
  • On p.2 line 51 “… underlying nurses’ decision-making process leading to the use of a physical restraint..” Suggest to discuss legal aspects of patient restraint in Italy. For instance, in some countries restraint must be prescribed by doctors or guided by established protocols. Is it the nurse who decide to use the physical restraint or the doctor has to prescribe the physical restraint? In terms of legal aspect, is it legal in Italy for physical restraint to be used for the convenience of caring, as mentioned on p.2 line 57 “context-related factors, such as the lack of human resources and the work environment quality”. Apart from legal implications, suggest to discuss the ethical implications.
  • On p.2 line 63 “…have investigated methods”. Suggest to discuss what do you mean by “methods”. Are you referring to the use of different physical restraint device or equipment?

Please clarify.

  • On p.2 line 90-93, suggest to review the research aim based on the legal implications as discussed earlier. If the use of physical restraint must be prescribed by doctor, then why interview nurses instead of doctor regarding decision making for the use of physical restraint? Or is the aim about the nurses’ perception of physical restraint use in the hospital setting and long term facilities?
  • On p.3 line 110, suggest to explain what do you mean by “the qualitative methods assumed a dominant status”? Consider elaborate on how does qualitative design address your research question?
  • On p.3 line 128, “…index day”, are you referring to “index date”? Suggest to explain what do you mean by “index day”
  • On p.3 line 129 and 130, suggest to clarify the inclusion and exclusion criteria for patient participants for phase 1 quantitate part of the study, and phase 2 qualitative part of the study. For instance, does the nurse need to care for patients with restraints or any patients appeared on the ward?
  • Suggest to discuss the psychometric properties of the “data collection tool” as mentioned on p.3 line 142.  Suggest to outline the steps for forward-backward translation.
  • On p.144 “Researcher accessed randomly the assigned units on randomised mornings”. Suggest to discuss the randomization techniques involved. Any considerations regarding the feasibility of the study, for instance, in acute care hospital units with deteriorating patients?
  • On p.3 line 148 “after receiving authorization”. Do you mean authorization from the hospital or long term facilities, or do you mean consent from patient?
  • On p.4 line 150 “in case of patient not able to give their consent, researchers awaited the hygiene care……and thus collect data”. Is it ethical to collect data from patient without obtaining their consent?
  • For table 1 on p.4, consider whether this is a quantitative study or whether it is a set of screening patients eligibility to fit the inclusion criteria. For instance, what are the dependent variables, independent variables or confounding variables in this study? Any deductive process or theory testing involved?
  • For table 1 on p.4, “one said bed rail + the other side near the wall”, suggest to modify “said” to “set”
  • For table 1 on p.4, review “Unilateral lower-limb wrist belt” and “Bilateral lower limb wrist belt”, do you mean “wrist” or “ankle”?
  • On p.4 line 165-166, please review the numbers “11 hospitals to 30 long term facilities”. These numbers are different when compared to 10 hospitals and 27 long term facilities as mentioned in the abstract.
  • On p.5 line 180 “involving all nurses”, do you mean nurse participants (90 as mentioned previously) ? Why there is “140 in the first and 30 in the second meeting”? Please review.
  • On p.5 line 209, have you collect any “sensitive information”? Please discuss.
  • On p.5 line 212, suggest to review the term “globally”. This study is conducted in Italy. Have data been collected from other countries?
  • Consider the sentence on p.5 line 253 …”team undertake the decision according to the needs expressed by relatives or by patients”, For Figure 1, how do we know relative’s concerns lead to preventing risk, not promoting comfort. Similarly, how do we kow patient’s need lead to promoting support but not preventing risk. Suggest to elaborate further.
  • On p.39, what does it mean by “sustaining posture was the intent”? Please explain or reword.
  • On Table 2, suggest to delete % on the row of Patients, n . Also, consider to rename this row as total number of patients present on hospital units and long term facilities
  • On Table 2, suggest to rename “Patients restrained” to “Patients restrained with restrictive device and supportive device”
  • On Table 2, Patients restrained row, long term care column, suggest to modifiy 1965 (49.9) to 1965 (50.0)

Author Response

Thank you for your comments, please see the attachment.

Round 2

Reviewer 2 Report

Thank you for giving me the opportunity to review the manuscript again. The authors have answered some of my previous comments, further clarification is suggested regarding some of my previous comments, including,

  1. “On p.1 line 17, “….= 4562 patients) were identified; then, for each patient, the nurse responsible was interviewed…”. Clarification is needed since p3 line 133 mentioned that only 90 nurses has participated in the study. Suggest to review the numbers.”

From my understanding, the study is a cross-sectional study, does a nurse usually take care of more than 50 patients in a shift on a particular day? 

  1. “On p.4 line 150 “in case of patient not able to give their consent, researchers awaited the hygiene care……and thus collect data”. Is it ethical to collect data from patient without obtaining their consent?”

Authors responded that “In light of the ethical issues…the researchers wear a uniform”. Please clarify whether the researchers are nurses or senior management working in the hospital? Any risk of coercion or conflict of interest?

  1. “For table 1 on p.4, consider whether this is a quantitative study or whether it is a set of screening patients’ eligibility to fit the inclusion criteria. For instance, what are the dependent variables, independent variables or confounding variables in this study? Any deductive process or theory testing involved?”

Apart from descriptive questions, the inferential questions that relate variables or compare groups appears to be missing.

Suggest to consider what is the purpose of mixing? How does the quantitative component and qualitative component integrate together?

Or, is this a qualitative study with inclusion criteria set to nurse caring for patient(s) with one or more restraint device?

Suggest to read:

Schoonenboom, J., & Johnson, R. B. (2017). How to construct a mixed methods research design. KZfSS Kölner Zeitschrift für Soziologie und Sozialpsychologie69(2), 107-131.

Based on this version,

On line 188, "second trained nurse researcher", does only one nurse researcher involved in the qualitative phase of the study? 

On line 336, “As reported in Table 2, No statistic”, suggest to change “No” to “no”

On table 2, please review “Patient with Ssupportive device(s), n (% patients) (% patients restrained)”

On line 355, suggest to change “in the attempt to” to “in attempt to”

On line 389-390, “collecting more data on clinical experience and knowledge of both participant nurses and researchers”. Please clarify or discuss the rationale of “collecting more data from researchers”?  

On line 415, suggest to delete “’: towards them,”

On line 512, please change “isto” to “is to”

On line 513, “not-restrictive tools”, do you mean “non-restrictive tools”? or do you  mean patients without restraints?

Author Response

Dear Reviewer, thank you for your further suggestion for improvement.

This was a mixed method study in which the first phase can be described as a cross sectional-study.

Only nurses in charge of restrained patients were interviewed. Thus, 66 nurses for 1965 patients in long-term care settings and 24 nurses for 268 in hospitals were interviewed. This means that in long-term care the nurse patient ratio was 1:30 and in hospital 1:11. These values mirror the Italian context meant as mean values among different wards.

We have specified this point better.

We clarified the point by highlighting the role of the researchers involved. No conflict of interest or coercion were experienced/applied.

Thank you for this point that helped to increase clarity.

Thank you, we have explained further this point in the data analysis section. Not all the outcomes (e.g., to be restrained with a supportive device) were definable a priori for the inferential analysis, given they are results of the data analysis, thus we have specified only the variables that were definable at the time of research protocol development.

Thank you for your precious suggestions. We have refined the method section according the article proposed.

We revised this section according to your suggestion.

Apologise for the inconsistencies

Corrected.

Round 3

Reviewer 2 Report

Thank you for addressing my comments.